# Regulation of Synaptic Development by Astrocyte Signaling Factors and Their Emerging Roles in Substance Abuse

**DOI:** 10.3390/cells9020297

**Published:** 2020-01-26

**Authors:** Christopher D. Walker, W. Christopher Risher, Mary-Louise Risher

**Affiliations:** 1Department of Biomedical Research, Joan C. Edwards School of Medicine, Huntington, WV 25701, USA; walker583@marshall.edu (C.D.W.); risherw@marshall.edu (W.C.R.); 2Neurobiology Research Laboratory, VA Medical Center, Huntington, WV 25704, USA; 3Department of Psychiatry and Behavioral Sciences, Duke University Medical Center, Durham, NC 27701, USA

**Keywords:** astrocytes, thrombospondin, netrin, apolipoprotein, neuregulins, bone morphogenetic proteins, neuroligin, alcohol use disorder, substance abuse, cocaine

## Abstract

Astrocytes have critical functions throughout the central nervous system (CNS) and have emerged as regulators of synaptic development and function. With their highly complex morphologies, they are able to interact with thousands of synapses via peripheral astrocytic processes (PAPs), ensheathing neuronal axons and dendrites to form the tripartite synapse. In this way, astrocytes engage in crosstalk with neurons to mediate a variety of CNS processes including the regulation of extracellular matrix protein signaling, formation and maintenance of the blood-brain barrier (BBB), axon growth and guidance, homeostasis of the synaptic microenvironment, synaptogenesis, and the promotion of synaptic diversity. In this review, we discuss several key astrocyte signaling factors (thrombospondins, netrins, apolipoproteins, neuregulins, bone morphogenetic proteins, and neuroligins) in the maintenance and regulation of synapse formation. We also explore how these astrocyte signaling factors are impacted by and contribute to substance abuse, particularly alcohol and cocaine use.

## 1. Introduction

In the last two decades, astrocytes have emerged as critical regulators of central nervous system (CNS) development and function. Astrocytes have complex morphologies with extensive peripheral astrocytic processes (PAPs) that extend from the soma to contact brain vasculature via endfeet [1] and ensheathe neuronal axon and dendrite compartments [2], forming the tripartite synapse. Each astrocyte can interact with up to 100,000 synapses in the rodent brain and an estimated 2 million synapses in humans [3], providing a wide-ranging network of connectivity that allows a single astrocyte to integrate and influence neuronal activity across independent circuits [4].

To date, it has been demonstrated that astrocytes regulate a variety of functions through contact-mediated signaling and secreted signaling factors. Astrocytes have been identified as key drivers in the formation of excitatory synapses in the CNS [5,6] and, more recently, in the signaling-dependent promotion of synaptic diversity [7]. Other significant astrocyte functions include: the regulation of extracellular matrix protein signaling [8,9], formation and maintenance of the blood-brain barrier (BBB) [10], neovascularization [11], neurogenesis [12], axonal growth [13], and homeostasis of the synaptic microenvironment [14].

A growing body of evidence suggests that astrocytes play important roles in cognitive impairment, neuronal loss, and synaptic deficits associated with aging and neuronal degenerative diseases [15,16]. In addition, astrocytes react to various conditions, including excitotoxicity, injury, age, and infection, in a process termed astrogliosis (i.e., astrocyte reactivity) [17]. Astrogliosis is a heterogeneous response in which these cells undergo context-dependent molecular and morphological changes. Depending on the nature of the insult, astrocytes can take on characteristic functional and molecular profiles termed A1 and A2 reactive astrocytes. The A1 reactive astrocyte subtype is characterized by the upregulation of complement 3 (C3, complement factor B (CFB)), and the MX Dynamin Like GTPase 1, MX1 [18]. The unique combination of microglia-secreted interleukin-1 alpha (IL-1α), tumor necrosis factor-alpha (TNFα), and complement component 1q (C1q) is required for the induction of A1 reactive astrocytes [18]. Following excitotoxicity, A1 reactive astrocytes contribute to neuronal and oligodendrocyte cell death to promote extensive remodeling of neuronal circuitry. If A1 astrocytes go unresolved and maintain their reactive phenotypes, they can have deleterious effects on synapses and subsequent neuronal function by inhibiting neuroplasticity and CNS regeneration [19]. By contrast, the A2 reactive astrocyte subtype is anti-inflammatory and neuroprotective. A2 reactive astrocytes can be induced following ischemia, which results in protracted upregulation of neurotrophic factors, promoting neuronal survival and synaptogenesis [18]. A2 reactive astrocytes are identified by the *S100A10* astrocyte-related gene, which is essential for cell proliferation, membrane repair, and inhibition of apoptosis [18]. A2 astrocytes promote the expression of the TNFα, which acts as an anti-inflammatory cytokine via the inhibition of inflammatory cytokines such as IL-12p40 [20], contributing to A2 astrocyte synaptogenetic and neuroprotective properties [18]. Both A1 and A2 reactive astrocytes are important in CNS recovery and restoration after injury or insult, but the specific pathways underlying A1 and A2 activation remain a major research focus as we currently do not know how, or if, there is a direct link between the two reactive subtypes.

Emergent evidence proposes that substances of abuse can result in reactive astrogliosis, including modification of astrocyte morphology and function through various processes, some of which involve plasmalemmal cholesterol composition remodeling [21,22]. In this regard, substance use can thereby disrupt neuronal circuits by triggering astrocytic changes similar to those observed in aging, injury, and disease models. This review will provide an overview of select emerging active astrocyte signaling factors (Figure 1) in the regulation of synaptic formation and maintenance. We are particularly focused on factors with known roles in synaptic connectivity and plasticity and those with potential involvement in substance abuse, specifically alcohol and cocaine use, that are just now beginning to be explored.

## 2. Thrombospondins

Thrombospondins (TSPs) were the first astrocytic factors shown to be necessary for the promotion of synaptogenesis within the CNS [6]. TSPs are a family of matricellular proteins that are an integral part of cell-to-cell communication [23,24]. There are five TSP isoforms expressed in mammals (TSP1-TSP5), all of which have been shown to be synaptogenic [6,25]. Though thrombospondin-1 (TSP1) and thrombospondin-2 (TSP2) promote excitatory synaptogenesis [6], they require other downstream signaling factors, including the neuronal receptor α2δ-1 [25] and the cytoskeletal regulator Rac1 [26], to promote the maturation and stabilization of the synapse. Through these neuronal factors, TSP signaling plays a significant role in dendritic spine maturation in the developing cortex [26]. It has also been demonstrated in TSP1 and TSP2 global knockout mice that TSPs are important in the maintenance of the neural progenitor pool and neuronal differentiation [27]. TSPs are found to be present at low concentrations, if at all, in adult astrocytes; however, expression levels of TSP1 and TSP2 are upregulated in response to injury [25]. These findings stress the importance of TSPs in the developing CNS and suggest their potential involvement in abnormal brain states.

In male rat models, TSP1 expression in the nucleus accumbens is elevated as a result of cocaine self-administration in adulthood [28]. TSP1 expression also remained elevated in this same region after extinction of cocaine [28]. These findings are significant for illustrating region-dependent changes to the reward pathway associated with addiction. In a model of adolescent binge drinking where male rats were administered intermittent ethanol via gavage, there is an acute upregulation of TSP2 observed 24 h after the last ethanol dose that persists into adulthood [29]. The increase in TSP2 is accompanied by protracted upregulation of thrombospondin-4 (TSP4) and astrocyte reactivity, which coincide with changes in astrocyte morphology. The acute TSP2 elevation in this study demonstrates an immediate response to adolescent intermittent ethanol exposure, while the chronic upregulation of TSP4 suggests that this response has longevity. These data suggest that adolescent intermittent ethanol exposure induces continued, aberrant synaptogenesis in the hippocampus that may contribute to the structural and functional abnormalities that strengthen the reward circuitry, with consequences well into adulthood.

## 3. Netrins

Netrins are a conserved family of laminin-related proteins that are expressed by oligodendrocytes and various endothelial cells throughout the body. Their receptors are highly expressed on the outer membrane of astrocytes [11,30]. Netrins act as multifunctional chemotrophic guidance cues for neuronal precursors and axonal projections during CNS development [30,31,32]. One of the best characterized netrins is netrin1, which is highly expressed throughout the developing human nervous system and is vital for neuronal survival and neovascularization of the brain [11,31,33]. Netrin1 guides axonal pathfinding by increasing axonal branching in the mammalian fetal cortex, contributing to the development of the human cerebral cortex [11,34]. In an adult male rodent model of stroke, netrin1 was found to be neuroprotective, promoting angiogenesis and anti-apoptotic activation [34]. Netrins also enhance synaptic regeneration of cortical neurons during postnatal synapse formation through the initiation of synaptic assembly [35]. Genetic global knockout of netrin has been shown to result in impaired hippocampal long-term potentiation (LTP) in adulthood (3–11 months of age) [36]. Taken together, these studies suggest that any disruption of netrin signaling, particularly during development, can have detrimental effects on neural survival, maturation, and/or plasticity.

The impact of alcohol on netrins in the CNS has yet to be thoroughly investigated. However, a recent study of cocaine dependency in a young adult male mouse (18–20 g) model (induced by daily intraperitoneal injections of cocaine [15 mg/kg]) revealed the downregulation of netrins and the impairment of netrin1-dependent axonal guidance signaling [37], thus disrupting the formation of synapses. A subsequent study using human brain samples from 34 to 54 year-old adults also showed downregulation of netrins in response to cocaine dependency [37]. In developmental rodent studies, loss-of-function or mutations of netrin1 or its receptors produced pups with impaired motor function; these pups would not suckle and died within a few days [32,38]. These studies highlight the importance of exploring the consequence of substances of abuse on netrins and their receptors not only in adult models but also during pregnancy and early development. This may lead to a further understanding of the role of astrocyte-specific factors in individuals born to parents with substance use disorders.

## 4. Apolipoproteins

Apolipoproteins are a family of proteins that function in the transport of lipids throughout the body, including the CNS [39]. The brain is the most lipid-rich organ, containing 20–25% of the total cholesterol in the human body [39,40]. Apolipoprotein E (APOE) is a primary cholesterol carrier that supports lipid transport and injury repair in the brain [41]. As a glycoprotein secreted primarily by astrocytes and, at much lower levels, neurons, APOE transports cholesterol between these cell types [42,43,44]. Disruption in cholesterol metabolism in the CNS can have serious effects on membrane fluidity and neurotransmitter function, leading to severe neurological disorders such as Lemli–Opitz syndrome, Niemann-Pick (type C, C1, and C2) disease, schizophrenia, Huntington’s disease, Parkinson’s disease, mood disorders, as well as unipolar and bipolar depression [45]. There are three *APOE* polymorphic alleles in humans (e2, e3, and e4) that are the leading genetic markers for Alzheimer’s disease (AD) [46,47]. The relationship between apolipoproteins and neurodegenerative disorders has been widely explored [48]. For example, it has been established that the presence of APOE4 alters the normal function of astrocytes, as well as other glial cell types, and may therefore represent a pathogenic mechanism that contributes to neurodegenerative pathways in AD and other disorders [46].

The relationship between apolipoproteins and addiction is a highly active area of investigation. A case study by Lewohl et al. compared adults who consumed an excess of 80 g of alcohol per day, which is defined as alcoholism by the National Health and Medical Research Council (NHMRC) and the World Health Organization, to social drinkers who consumed less than 20 g on average per day. This study revealed that alcohol reduces *APOE* mRNA expression in the prefrontal cortex (PFC) [49], contributing to the disruption of cholesterol homeostasis in the CNS [50]. This impairment was characterized by abnormal membrane fluidity and neurotransmitter release and uptake by both neurons and astrocytes from the PFC. A population-based prospective study reported a correlation between carriers of the *APOE4* allele over the age of 65 who consumed alcohol and increased likelihood of ischemic stroke [51]. Furthermore, another cross-sectional study of AD patients demonstrated a correlation among *APOE4*-positive individuals, heavy alcohol consumption, heavy tobacco use, and AD [52]. There has also been a correlation found between the extinction of cigarette smoking and elevated APOE4 expression [50]. Taken together, these studies suggest that heavy alcohol consumption and cigarette smoking, which are typically co-morbidly expressed, increase the likelihood of AD and ischemic stroke. Further risk of these diseases is observed in individuals who present with *APOE4*. Moreover, onset of AD was 2–3 years earlier in carriers of the *APOE4* allele who had a history of heavy alcohol use (more than two drinks per day) or heavy tobacco use (a pack or more of cigarettes per day) than for non-*APOE4* controls [52]. In *APOE4*-positive individuals who were both heavy drinkers and smokers, the onset of AD was up to 10 years earlier than for those with none of these risk factors [52]. There are currently no studies that have correlated cocaine use and abuse with the presence of *APOE4*; however, using a condition place preference (CPP) assay, Bechtholt, Smith et al. (2004) observed that cocaine does not activate the reward pathway in the same manner as EtOH in global APOE knockout mice during adolescence and emerging adult (ages 2–5 months) in males and females [50]. While more studies are needed, these data would suggest that APOE4 may have a unique role in the abuse of alcohol but not cocaine.

## 5. Neuregulins

Neuregulins (NRGs) are cell-to-cell signaling proteins expressed by astrocytes and neurons in the CNS, interacting indirectly with tyrosine kinases of the ErbB family to promote cell survival, proliferation and differentiation [53]. NRG isoforms with a cysteine-rich domain in the N-terminus serve to regulate nicotinic acetylcholine receptors (nAChRs) during synaptogenesis [54]. Most NRG research to date has focused on astrocyte expression of neuregulin-1 and its association with neuropsychiatric disease. Neuregulin-1 (NRG1) signaling is thought to play a role in the development and function of normal neuronal connectivity in the CNS and peripheral nervous system as well as in other organs outside of the nervous system [55]. Inhibition of NRG1 signaling has been implicated in numerous neuropsychiatric disorders such as schizophrenia, AD, and bipolar disorder [55]. Studies using rodent cortical neurons have shown a link between inhibition of the NRG1-ErB4 signaling pathway and the disruption of synaptic transmission, myelination, and neuronal survival in cortical neurons [55], which may contribute to the development of these disorders. NRG1 is further divided into three major families (types I, II, and III) and three minor families (IV, V, and VI) [53,55]. Studies using transgenic overexpressing NRG1 type 1 (*Nrg1*^tg-type1^) adult male and female mice have shown altered gene expression and structure in the hippocampus, which correlates with immune function and symptoms resembling those characteristic of schizophrenia [56]. Furthermore, overexpression of NRG1 type I results in deficits in hippocampus-dependent behavior such as hyperactivity and deficits in spatial working memory [57]. This shift in normal hippocampus-dependent behavior to ‘schizophrenia-like’ behavior is likely due to the disruption of neural circuitry resulting in increased GABA concentrations in the synaptic microenvironment [57].

The roles of NRG in immune function, neurogenesis, and synaptogenesis demonstrate functional overlap with a number of critical events related to the development of addiction. Interestingly, a number of the aforementioned neuropsychiatric disorders are associated with higher levels of alcohol use disorder (AUD) and substance use disorder (SUD), suggesting that NRG1 type I isoform would be a promising mechanistic target for investigation into cocaine and alcohol addiction. A longitudinal population study by Vaht et al. [58] provided further support of a potential role for NRG1 in addiction; it revealed an increased risk of alcohol and substance abuse in humans with the *NRG1* genotype that also correlated with a high number of adverse life events when compared to populations in which these events were minimal. This finding suggests that NRG1 expression is not only associated with substance abuse, but that its expression can be moderated by stressful life events. To date, work in animal models has been limited. Overexpression of the NRG type 1 isoform in a mouse model decreases activity in the hippocampus, disrupting synaptogenesis and transmission of signals in cortical interneurons [56,57]. This resultant decrease in adult hippocampal activity was shown to contribute to the disruption of spatial memory in a spatial nonmatching-to-place T-maze test [57]. These data suggest that changes in NRG1 type I expression could contribute to the development of substance abuse, particularly in the presence of stress, or that substance abuse and stress upregulate NRG1 type I with potential detrimental effects to cognitive function. The disruption of hippocampus-facilitated spatial working memory contributes to the ‘schizophrenia-like’ behavior previously observed. Further work is necessary to delineate this relationship among NRG1 type I, stress, and the development of neuropsychiatric disorders, as well as to determine whether NRG type 1-positive individuals are more prone to alcohol and substance abuse.

## 6. Bone Morphogenetic Proteins

Bone morphogenetic proteins (BMPs) are the largest subclass of multifunctional growth factors of the transforming growth factor-beta (TGF-β) superfamily, with 20 structurally distinct members [59,60,61,62]. BMPs act in the regulation of bone induction, repair, and maintenance [63]. In the CNS, BMPs secreted by astrocytes have been found to play an essential role in the maintenance of homeostasis of the synaptic microenvironment [60,61]. BMPs have also been found to function as astrocyte differentiation factors as well as regulators for cell adhesion molecule expression in the developing nervous system [64]. In particular, studies have shown that BMP4 functions in the regulation of neuronal and glial cell development from neuronal precursor cells during embryonic and postnatal development as well as in the presence of neuronal damage [62]. BMPs are especially significant in prenatal development as they promote embryonic neuroectoderm induction, neural crest specification, and CNS neuronal patterning [62]. Taken together, these findings suggest that disruption of BMPs can lead to impaired prenatal development as well as subsequent impairments to neuronal and glial function.

Changes in the expression of BMPs, as can occur with exposure to substances of abuse, can have deleterious effects on the brain [64]. In cell culture studies, it has been found that the presence of EtOH inhibits BMP2, BMP4, BMP5, and BMP6-induced cell adhesion, altering nervous system development [65]. In conjunction with other growth factors such as insulin-like growth factor 1 (IGF-1), BMPs have been found to regulate N-methyl-D-aspartate type glutamate receptor (NMDAR) function [66]. This finding is significant because it has been well-established that NMDA receptors are a major target for alcohol and cocaine and play a prominent role in the reward pathway, associated with tolerance, dependence, withdrawal, cravings, and relapse [67]. Regulation of NMDAR signaling by BMPs may therefore have widespread consequences for the CNS, with hypofunction of NMDAR signaling associated with memory and learning impairments, psychosis, and excitotoxic brain injury, while NMDAR hyperfunction has been implicated in the pathophysiology of acute CNS injuries such as hypoxia-ischemia, trauma, and epilepsy [68]. To date, there are limited data on the direct role of BMPs in cocaine addiction. However, one study that used a human cell culture model demonstrated that the use of cocaine in the presence of HIV-1 downregulates BMP receptors [69]. Based on this knowledge, we speculate that the disruption of BMPs by alcohol and/or drugs of abuse may perturb downstream NMDA receptor function and thereby contribute to alcohol and/or substance abuse and addiction.

## 7. Neuroligins

Neuroligins are cell adhesion molecules highly expressed by neurons and, as was recently demonstrated, astrocytes [70,71]. There are three types of neuroligins (NL1, NL2, and NL3), whose expression has been found to be comparable or even greater in astrocytes when compared to neurons [71,72]. These neuroligins are expressed at the terminal ends of PAPs, where they interact with neuronal and synaptic proteins [71], and where they play important roles in contact-mediated signaling. NL1 participates in the recruitment of receptors, channels, and signal-transduction of molecules to the excitatory synapse [72]. NL2 is important in the formation of inhibitory synaptogenesis and is localized to the postsynaptic membrane [70], while NL3 localizes to both glutamatergic and GABAergic synapses [73]. It has been demonstrated in C57B1/6N mice that, in the cortex, NL1, NL2, and NL3 are involved in astrocyte morphogenesis and the stabilization of synapses through their interactions with neurexins on the presynaptic terminus [71,74]. In rodent cell cultures, hevin (SPARC-like 1), an astrocyte-secreted synaptogenic protein, facilitates the bridging of incompatible neurexin and neuroligin pairs to assemble and stabilize synapses [75]. Disruption of the neuroligin–neurexin relationship has been associated with autism and other neurological diseases [76], leading some groups to speculate whether these interactions are perturbed in the pathology of addiction. 

While the direct consequences of substances of abuse on neuroligins are still largely unknown, neuroligin interactions with their targeted neurexin (NRXN) receptors have been documented in abuse models. Human genomic studies of neurexin 3 (NRXN3) have shown disruption of brain structure, function, and subsequent behavior with nicotine, opioid, and alcohol addictions through the alteration of expression of NRXN3 isoforms [77]. By disrupting the bridging of astrocytic NL and NRXN on the pre- or postsynaptic terminals, the stabilization of the tripartite synapse is compromised [75], preventing the proper astrocyte-neuronal crosstalk and ensheathment of the synapse by the astrocyte. This disruption would then provide a mechanism for synaptic dysfunction with drugs of abuse, since excess glutamate neurotransmitters are removed from the synapse via astrocyte specific Na+-dependent glutamate transporters, GLT-1 and GLAST [78,79]. Should the tripartite synapse not be properly assembled or maintained, the resulting increased proximal distance of the astrocytic processes to synapses may allow excess glutamate to accumulate and contribute to neuronal hyperexcitability, a hallmark of alcohol and substance abuse [80]. These studies further stress the importance of studying not only the effects of substances of abuse on the neuroligin–neurexin interaction but also the effects on astrocyte-secreted neuroligins more specifically.

## 8. Conclusions

A growing body of evidence highlights the critical contributions of astrocytes to normal CNS function, including synapse formation, maintenance, transmission, and plasticity. Through the study of astrocyte activity in drug and alcohol models, we may further understand the effects of substance abuse on the synaptic circuitry of the reward pathway. Here, we have considered the normal function of various astrocyte-enriched factors and discussed how they contribute to neuronal deficits in the presence of cocaine and alcohol. Further study of these astrocyte-specific targets may lead to novel, innovative therapeutic applications to combat drug and alcohol abuse.

## Figures and Tables

**Figure 1 cells-09-00297-f001:**
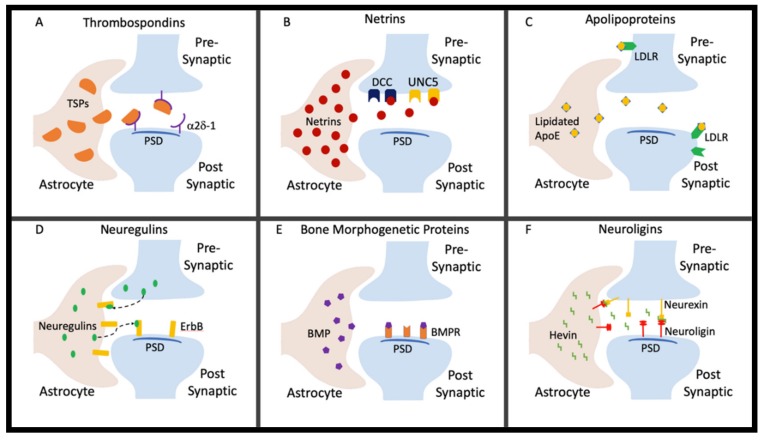
Astrocytes secrete factors that regulate extracellular matrix protein signaling, formation and maintenance of the blood-brain barrier (BBB), axonal growth, homeostasis of the synaptic microenvironment, synaptogenesis, and the promotion of synaptic diversity. (**A**) Thrombospondins bind to the neuronal receptor α2δ-1 at the postsynaptic density to form a silent synapse. (**B**) Netrins bind to multiple receptors on the presynaptic terminus such as Deleted in Colorectal Cancer (DCC) and UNC-5. (**C**) Apolipoproteins containing cholesterol and lipids are secreted from the astrocyte and bind to the receptors on the presynaptic and postsynaptic shafts. (**D**) Neuregulins are secreted by the presynaptic terminus and, in higher concentrations, astrocytes. Neuregulins secreted by the presynaptic terminus bind to ErbB receptors on the astrocyte, while neuregulins secreted by astrocytes bind to ErbB receptors on the postsynaptic terminus. (**E**) Bone morphogenetic proteins are secreted by astrocytes and bind to bone morphogenetic protein receptors (BMPRs) at the postsynaptic density. (**F**) Neuroligins located postsynaptically and on astrocytes bind neurexins on the presynaptic terminus directly and indirectly. Incompatible proteins are bridged by astrocyte-secreted hevin to stabilize the synapse.

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
