# Peer review of "Regulation of Synaptic Development by Astrocyte Signaling Factors and Their Emerging Roles in Substance Abuse"

_cells, 2020, doi:10.3390/cells9020297_

Round 1

Reviewer 1 Report

In this review entitled, "Regulation of synaptic development by astrocyte signaling factors and their emerging roles in substance abuse", Walker et al. review a number of key molecules and receptors secreted and/or expressed by astrocytes that play an important role in development of mature synapses. The authors further discuss the potential roles of these factors in the development or consequence of substance abuse. Overall the manuscript is very well-written and organized. After a general introduction, each subheading is by factor/protein type with a paragraph on their role in development followed by a paragraph reviewing previous work on their alterations or effects on substance abuse. It is easy to read and provides an original contribution to the field. There is one more prominent, plus a couple of other relatively minor revisions that should be considered prior to publication. 

Major:

In the sections on substance abuse, it is often difficult to determine if the authors are referring to findings or studies in adults or in juvenile/adolescents. This should be clarified as it greatly effects the interpretation of the results, as many of the factors and proteins are expressed at high levels during development but presumably reduce their expression with maturation. In some instances, it seems that the work is from exposure during fetal development or early postnatal development via suckling (such as ethanol studies), while in other cases it seems that the findings were obtained from mature animals or from examination of adult human tissues (such as effects of cocaine use). The point is, it's very difficult to tell. For example, in the netrins section, starting from line 126 a study of cocaine dependency is discussed, mentioning that downregulation of netrins and and netrin-1 dependent signaling disrupts formation of synapses. Was the cocaine therefore administered to pregnant dams or young postnatal animals? In general, findings could be completely different in juveniles vs. adults due to the important and essential role these factors play during development. During development it could lead or contribute to neurodevelopmental disorders, while in the adult changes could lead to alteration of synapses contributing to addiction. The manuscript would be greatly strengthened by making this more clear in each instance. 

Minor:

On p. 2 line 49 it says that "A1 reactive astrocytes are induced by microglial activation and the secretion of... TNF-alpha". Then on line 59 the authors state that "A2 astrocytes promote the expression of the anti-inflammatory cytokine TNF-alpha". This is very confusing. How can TNF-alpha be anti-inflammatory if it contributes to the development of the A1 inflammatory reactive astrocyte subtype?  On p. 6 line 236, the authors state, "This finding is significant because, as previously discussed, it is well-established that NMDA receptors are a major target for alcohol and cocaine". This was not previously discussed. 

Author Response

We would like to thank the reviewers for their insightful and constructive feedback. We appreciate the time and effort the reviewers have dedicated to provide valuable commentary on our review. We have been able to incorporate changes to our manuscript that reflect the feedback provided by the reviewers. All changes from the original text have been changed to blue font.

Reviewer #1:

In the sections on substance abuse, it is often difficult to determine if the authors are referring to findings or studies in adults or in juvenile/adolescents. This should be clarified as it greatly effects the interpretation of the results, as many of the factors and proteins are expressed at high levels during development but presumably reduce their expression with maturation. In some instances, it seems that the work is from exposure during fetal development or early postnatal development via suckling (such as ethanol studies), while in other cases it seems that the findings were obtained from mature animals or from examination of adult human tissues (such as effects of cocaine use). The point is, it's very difficult to tell. For example, in the netrins section, starting from line 126 a study of cocaine dependency is discussed, mentioning that downregulation of netrins and and netrin-1 dependent signaling disrupts formation of synapses. Was the cocaine therefore administered to pregnant dams or young postnatal animals? In general, findings could be completely different in juveniles vs. adults due to the important and essential role these factors play during development. During development it could lead or contribute to neurodevelopmental disorders, while in the adult changes could lead to alteration of synapses contributing to addiction. The manuscript would be greatly strengthened by making this more clear in each instance.

            We agree that substances of abuse can have varying effects on neuronal function, depending on the time in which the drug was introduced. When writing this review, we took an approach that would allow us to discuss alcohol and cocaine use during developmental periods as well as adulthood and how adverse effects on astrocyte secreted factors result in neuronal impairment can arise, though we now realize that we did not make the age distinctions appropriately throughout the text. We have reviewed the manuscript and changed the language to better describe the life-stages and sex of the animal models used in the cited studies.

On p. 2 line 49 it says that “A1 reactive astrocytes are induced by microglial activation and the secretion of… TNF-alpha”. Then on line 59 the authors state that “A2 astrocytes promote the expression of the anti-inflammatory cytokine TNF-alpha” This is very confusing. How can TNF-alpha be anti-inflammatory if it contributes to the development of the A1 inflammatory reactive astrocyte subtype?

            We apologize for the confusion. Previous studies, such as the one referenced, have determined that the unique combination of activated microglia secreted cytokines IL-1a, TNFa, and C1q are required for A1 astrocyte activation. While TNFa is anti-inflammatory, its presence with the previous mentioned cytokines activate A1 astrocytes, which can be detrimental if left unresolved. A2 astrocyte induction promotes the upregulation of neuroprotective factors, including TNFa, to suppress inflammation. We have addressed this on page 2, lines 59-62. Currently we do not know the specific pathways that underly the activation A1 and A2 astrocytes (lines 62-65) or if their activation is related (i.e. if A1 activation induces A2 activation or vice-versa). This uncertainty highlights the importance of further investigation into these astrocyte activation pathways.

On p. 6 line 236, the authors state “This finding is significant because, as previously discussed, it is well-established that NMDA receptors are a major target for alcohol and cocaine”. This was not previously discussed.  

            Thank you for bringing this to our attention. Through numerous edits and restructuring of our manuscript prior to submission, the information concerning NMDA receptors had been rearranged into this one area. We have removed “as previously discussed” to better reflect the current organization of the manuscript.

Reviewer 2 Report

Review: ‘Regulation of synaptic development by astrocyte signalling factors and their emerging roles in substance abuse’ by Walker et al.,

In this review, Walker and colleagues describe a variety of signalling factors released by astrocytes (e.g. thrombospondins, netrins and neuroligins) and how these factors are impacted by substance abuse and may even contribute to it.

The authors address a highly interesting topic; the review is clearly structured and well-written. The authors provide a comprehensive summary how thrombospondins, netrins, apolipoproteins, and others released by astrocytes contribute to synaptogenesis, neuronal circuits and astrogliosis. Furthermore, they tackle the important issue how the release of these substances is changed in mouse models of substance abuse, in particular for alcohol and cocaine.

In general the review is of interest for the readership of Cells. Yet, minor changes would be a welcome addition before a publication.

Major points:

It remains unclear, at least to the non-specialist, why the authors have chosen certain signalling factors, but not others (e.g. glutamate or interleukins). In this context, it would also be highly interesting to discuss how gliotransmitter release is altered by substance abuse. For example, Turner and colleagues described that a dominant-negative (dn)- SNARE mouse model (astrocytic gliotransmission is blocked) do not demonstrate cue-induced reinstatement of cocaine self-administration. Similarly conditioned place preference was also blocked in dnSNARE mice (Turner et al., 2013). The authors describe the role of Thrombospondins TSP1 and TSP2 in astrocytes, but in line 105, they mention the protracted upregulation of TSP4. For the sake of clarity, the function of TSP4 should also be introduced better. For non-specialists, results from experiments or studies in genetically modified mouse models should specify whether an astrocyte-specific or a global mouse model was employed. This is particularly important for studies of apolipoprotein and neuroligin, which are both also expressed in neurons.

Minor points:

Line 53 ‘phonotypes’ probably means ‘phenotypes’.   Lines 165-167 are confusing and should be reformulated

Author Response

We would like to thank the reviewers for their insightful and constructive feedback. We appreciate the time and effort the reviewers have dedicated to provide valuable commentary on our review. We have been able to incorporate changes to our manuscript that reflect the feedback provided by the reviewers. All changes from the original text have been changed to blue font.

Reviewer #2:

It remains unclear, at least to the non-specialist, why the authors have chosen certain signalling factors, but not others (e.g. glutamate or interleukins). In this context, it would also be highly interesting to discuss how gliotransmitter release is altered by substance abuse. For example, Turner and colleagues described that a dominant-negative (dn)- SNARE mouse model (astrocytic gliotransmission is blocked) do not demonstrate cue-induced reinstatement of cocaine self-administration. Similarly conditioned place preference was also blocked in dnSNARE mice (Turner et al., 2013).

            There is a growing library of astrocyte signaling factors that play important roles in the regulation of synaptic formation, maintenance, and signal transmission. Therefore, the factors chosen for the current review are far from an exhaustive list. For this review we have focused on astrocyte-specific factors that are just now beginning to be explored in these substance use models. On Page 2, lines 69-73, we reworded the summation of the review to state, “This review will provide an overview of select emerging active astrocyte signaling factors in the regulation of synaptic formation and maintenance. We are particularly focused on factors with known roles in synaptic connectivity and plasticity and those with potential involvement in substance abuse, specifically alcohol and cocaine use, that are just now beginning to be explored.”

The authors describe the role of Thrombospondins TSP1 and TSP2 in astrocytes, but in line 105, they mention the protracted upregulation of TSP4. For the sake of clarity, the function of TSP4 should also be introduced better.

            The manuscript has been edited to include a statement about the synaptogenic capabilities of all 5 mammalian TSP isoforms, including TSP4 on page 3, lines 92-94.

For non-specialists, results from experiments or studies in genetically modified mouse models should specify whether an astrocyte-specific or a global mouse model was employed. This is particularly important for studies of apolipoprotein and neuroligin, which are both also expressed in neurons.         

We have read through the manuscript and included specifics regarding genetic manipulations of the animal models, ages, and sex. These changes can be found throughout the revised manuscript: page 3 (line 98-99), page 4 (line 128), and page 5 (line 183),

Line 53 ‘phonotypes’ probably means ‘phenotypes’.

            Thank you for bringing this to our attention. This was a typo that has now been corrected.

Lines 165-167 are confusing and should be reformulated.

            Thank you for your feedback. Upon further reading, we agree these statements could have been stated more articulately. We have reworded this section accordingly.

Reviewer 3 Report

The authors summarized several astrocytic factors that affect synaptic formation and maintenance. Also, they provided detailed examples how those factors have a role in substance abuse. This is a very interesting and well summarized review paper. The present manuscript is of sufficient impact and general interest to warrant publications in cells. However, further discussion may be needed to address certain concerns in the manuscript.

Further descriptions about signal transmissions should be provided in details and with appropriate references. The authors stated that this review would provide the role of astrocyte signaling in the regulation of synaptic formation maintenance, and “signal transmission”. However, most contents are focused on how the factors secreted from astrocytes form and maintain synapses. There are just few descriptions showing the effects in the neurotransmitter by the factors secreted from astrocytes without details and appropriate references, although they beautifully described the localization and interaction of target molecules and receptors in the figure 1. For example, the authors described that NRG1 overexpression leading schizophrenic-like behavior is likely due to increased GABA concentrations [1]. However, no data to talk about GABA concentration was provided in the reference [1]. Instead of this, there are other previous studies showing the direct GABA-related electrophysiological synaptic properties which are modulated by Neuregulin 1-ErbB4 signaling [2,3]. In addition, other neurotransmitters including glutamate are also suggested to be modulated by the factors [4,5]. Provide the role of those factors in modulation of neurotransmitters specifically. The authors introduced two major different astrocytes, A1 and A2 reactive astrocytes, well in the introduction. Provide any discussion for the significant relation between those specific astrocytic types and the described astrocytic factors. Some of the citations are not appropriate. For example, the articles suggesting the role of netrin1-mediated axonal elongation with alcohol conc. of 25 mM has been retracted [6]. Also, this in vitro experiment even was not described clearly. Since this manuscript is a review article, I recommend to avoid citing retracted articles or to provide a clear statement about the retraction. Please provide descriptions clearly with appropriate references. Please increase the font size of the texts in the figure.

[1] Deakin, I.H.; Nissen, W.; Law, A.J.; Lane, T.; Kanso, R.; Schwab, M.H.; Nave, K.A.; Lamsa, K.P.; Paulsen, O.; Bannerman, D.M., et al. Transgenic overexpression of the type I isoform of neuregulin 1 affects working memory and hippocampal oscillations but not long-term potentiation. Cereb Cortex 2012, 22, 1520-1529, doi:10.1093/cercor/bhr223.

[2] Woo, R.S.; Li, X.M.; Tao, Y.; Carpenter-Hyland, E.; Huang, Y.Z.; Weber, J.; Neiswender, H.; Dong, X.P.; Wu, J.; Gassmann, M., et al. Neuregulin-1 enhances depolarization-induced GABA release. Neuron 2007, 54, 599-610, doi:10.1016/j.neuron.2007.04.009.

[3] Geng, F.; Zhang, J.; Wu, J.L.; Zou, W.J.; Liang, Z.P.; Bi, L.L.; Liu, J.H.; Kong, Y.; Huang, C.Q.; Li, X.W., et al. Neuregulin 1-ErbB4 signaling in the bed nucleus of the stria terminalis regulates anxiety-like behavior. Neuroscience 2016, 329, 182-192, doi:10.1016/j.neuroscience.2016.05.018.

[4] Li, B.; Woo, R.S.; Mei, L.; Malinow, R. The neuregulin-1 receptor erbB4 controls glutamatergic synapse maturation and plasticity. Neuron 2007, 54, 583-597, doi:10.1016/j.neuron.2007.03.028.

[5] Yu, H.N.; Park, W.K.; Nam, K.H.; Song, D.Y.; Kim, H.S.; Baik, T.K.; Woo, R.S. Neuregulin 1 Controls Glutamate Uptake by Up-regulating Excitatory Amino Acid Carrier 1 (EAAC1). J Biol Chem 2015, 290, 20233-20244, doi:10.1074/jbc.M114.591867.

[6] Chen, S.; Charness, M.E. Ethanol disrupts axon outgrowth stimulated by netrin-1, GDNF, and L1 by blocking their convergent activation of Src family kinase signaling. J Neurochem 2012, 123, 602-612, doi:10.1111/j.1471-4159.2012.07954.x.

Author Response

We would like to thank the reviewers for their insightful and constructive feedback. We appreciate the time and effort the reviewers have dedicated to provide valuable commentary on our review. We have been able to incorporate changes to our manuscript that reflect the feedback provided by the reviewers. All changes from the original text have been changed to blue font.

Reviewer #3:

Further descriptions about signal transmissions should be provided in details and with appropriate references. The authors stated that this review would provide the role of astrocyte signaling in the regulation of synaptic formation maintenance, and “signal transmission”. However, most contents are focused on how the factors secreted from astrocytes form and maintain synapses. There are just few descriptions showing the effects in the neurotransmitter by the factors secreted from astrocytes without details and appropriate references, although they beautifully described the localization and interaction of target molecules and receptors in the figure 1. For example, the authors described that NRG1 overexpression leading schizophrenic-like behavior is likely due to increased GABA concentrations [1]. However, no data to talk about GABA concentration was provided in the reference [1]. Instead of this, there are other previous studies showing the direct GABA-related electrophysiological synaptic properties which are modulated by Neuregulin 1-ErbB4 signaling [2,3]. In addition, other neurotransmitters including glutamate are also suggested to be modulated by the factors [4,5]. Provide the role of those factors in modulation of neurotransmitters specifically.

            Thank you for bringing this to our attention. We believe these astrocyte-specific factors are important in signal transmission and agree that we did not explore their roles in signaling to the extent necessary. Due to the time constraint for revisions we have decided to remove ‘signal transmission’ from the abstract on page 1 (line 20) and page 2 (line 69-73) to reflect the theme of the manuscript which focuses on synaptic formation and maintenance.

The authors introduced two major different astrocytes, A1 and A2 reactive astrocytes, well in the introduction. Provide any discussion for the significant relation between those specific astrocytic types and the described astrocytic factors.

            We discussed the differences between A1 and A2 reactive astrocytes because they are important in the response to injury and insult. Our lab is very interested in this relationship and how factors, such as those discussed within the review, are associated. However, at this time there is a gap in the understanding of the relationship between A1 and A2 astrocytes. This is reflected in the additional text that has been added (page 1, lines 62-65).

Some of the citations are not appropriate. For example, the articles suggesting the role of netrin1-mediated axonal elongation with alcohol conc. of 25 mM has been retracted [6]. Also, this in vitro experiment even was not described clearly. Since this manuscript is a review article, I recommend to avoid citing retracted articles or to provide a clear statement about the retraction. Please provide descriptions clearly with appropriate references.

            Thank you for pointing out the retracted article. We were unable to determine the reason(s) for the retraction and therefore removed it from our review.

Please increase the font size of the texts in the figure.

            We have re-worked our image so that the text is larger and more legible.